# Oligonol^®^, an Oligomerized Polyphenol from *Litchi chinensis*, Enhances Branched-Chain Amino Acid Transportation and Catabolism to Alleviate Sarcopenia

**DOI:** 10.3390/ijms252111549

**Published:** 2024-10-27

**Authors:** Yun-Ching Chang, Yu-Chi Chen, Yin-Ching Chan, Cheng Liu, Sue-Joan Chang

**Affiliations:** 1School of Medicine, College of Medicine, I-Shou University, Kaohsiung 82445, Taiwan; ychang014@isu.edu.tw (Y.-C.C.);; 2Department of Urology, E-Da Cancer Hospital, I-Shou University, Kaohsiung 82445, Taiwan; 3Department of Food and Nutrition, Providence University, Taichung 43330, Taiwan; ycchan@pu.edu.tw; 4Department of Physical Therapy, Shu-Zen Junior College of Medicine and Management, Kaohsiung 82144, Taiwan; 5Department of Life Sciences, National Cheng Kung University, Tainan 701, Taiwan; 6Marine Biology and Cetacean Research Center, National Cheng Kung University, Tainan 701, Taiwan

**Keywords:** sarcopenia, oligonol^®^, low-molecular-weight polyphenol, branched-chain amino acids, L-type amino acid transporter 1, branched-chain amino acid transaminase 2, protein synthesis

## Abstract

Branched-chain amino acids (BCAAs) are essential for muscle protein synthesis and are widely acknowledged for mitigating sarcopenia. Oligonol^®^ (Olg), a low-molecular-weight polyphenol from *Litchi chinensis*, has also been found to attenuate sarcopenia by improving mitochondrial quality and positive protein turnover. This study aims to investigate the effect of Olg on BCAA-stimulated protein synthesis in sarcopenia. In sarcopenic C57BL/6 mice and senescence-accelerated mouse-prone 8 (SAMP8) mice, BCAAs were significantly decreased in skeletal muscle but increased in blood serum. Furthermore, the expressions of membrane L-type amino acid transporter 1 (LAT1) and branched-chain amino acid transaminase 2 (BCAT2) in skeletal muscle were lower in aged mice than in young mice. The administration of Olg for 8 weeks significantly increased the expressions of membrane LAT1 and BCAT2 in the skeletal muscle when compared with non-treated SAMP8 mice. We further found that BCAA deprivation via LAT1-siRNA in C2C12 myotubes inhibited the signaling of protein synthesis and facilitated ubiquitination degradation of BCAT2. In C2C12 cells mimicking sarcopenia, Olg combined with BCAA supplementation enhanced mTOR/p70S6K activity more than BCAA alone. However, blocked LAT1 by JPH203 reversed the synergistic effect of the combination of Olg and BCAAs. Taken together, changes in LAT1 and BCAT2 during aging profoundly alter BCAA availability and nutrient signaling in aged mice. Olg increases BCAA-stimulated protein synthesis via modulating BCAA transportation and BCAA catabolism. Combining Olg and BCAAs may be a useful nutritional strategy for alleviating sarcopenia.

## 1. Introduction

Sarcopenia is a progressive skeletal muscle disorder that accrues across a lifetime, especially in the elderly population. The European Working Group on Sarcopenia in Older People 2 (EWGSOP2) advised in 2018 that low muscle strength should serve as the principal criterion for diagnosing sarcopenia, with the diagnosis being substantiated by further evidence of reduced muscle quantity or quality [1]. Sarcopenia impairs activities of daily living, leads to poor life quality, increases the risk of falls and fractures, and is a poor prognostic factor for many diseases such as chronic obstructive pulmonary disease (COPD), diabetes mellitus, cardiovascular diseases, and cancer [2]. The prevalence of sarcopenia in Taiwan has been reported as ranging from 6.7% to 10% in communities and more than 50% in clinical settings [3,4,5]. Sarcopenia results in a heavy public health burden and heavy expenditure in healthcare. A 10% reduction in the incidence of sarcopenia could result in annual healthcare savings of USD 1.1 billion by reducing the need for surgeries and hospitalizations among patients [6]. Therefore, preventing and alleviating sarcopenia is vital for improving life quality and survival rates in the elderly and reducing public health burdens and expenditures in healthcare.

Exercise positively affects muscle mass, strength, and physical function and is considered the best way to prevent sarcopenia [7]. Nevertheless, elderly individuals, particularly those who are physically vulnerable, frequently encounter difficulties in maintaining an exercise routine. In addition to exercise, dietary and nutritional interventions may be an effective strategy for preventing or attenuating sarcopenia. Recent research has indicated that the supplementation of protein and amino acids, whether combined with exercise or not, improves muscle mass and strength while also maintaining physical performance and function in pre-frail older adults. [8]. Twenty amino acids, including nine essential amino acids, compose muscle protein and contribute to protein synthesis to replace the protein lost due to protein breakdown. Moreover, amino acids act as signaling molecules in addition to being building blocks for muscle proteins. It has been widely accepted that branched-chain amino acids (BCAAs: leucine, valine, and isoleucine) are not only used as substrates for protein synthesis but also act as signaling molecules to stimulate skeletal muscle protein synthesis [9]. As an activator of the mammalian target of rapamycin (mTOR), leucine promotes protein synthesis via phosphorylating the downstream of mTOR, including ribosomal protein S6 kinase (S6K1) and eukaryotic mRNA translation initiation factor 4E-binding protein 1 (4E-BP1) [9,10]. Valine protects against muscle damage induced by oxidative stress via the up-regulation of genes involved in mitochondrial biogenesis such as PPAR-gamma coactivator-1 (PGC-1) [11]. Isoleucine increases muscle mass through stimulation of myogenesis-associated proteins such as myosin heavy chain (MyHC) and myoblast determination protein 1 (MyoD) [12]. Although BCAAs are highly effective at growing muscle, excessive intake may lead to a shortened lifespan, hyperphagia, weight gain, and a negative impact on mood [13]. Indeed, the beneficial effect of BCAA supplementation on sarcopenia is also controversial.

The molecular mechanisms of sarcopenia are not entirely understood, but the age-associated accumulation of oxidative stress and inflammation in skeletal muscle cells is one of the most accepted underlying pathways [14]. Accordingly, numerous nutraceuticals and functional foods are well known for their antioxidant and anti-inflammatory properties and gain special interest in aging-associated diseases. Oligonol^®^ (Olg) is composed mainly of lychee polyphenols and is especially rich in proanthocyanidin-derived monomers (15–20%), dimers (8–12%), and oligomers (5–10%) [15]. The major phenolic contents of Olg are epicatechin (7.5%), epicatechin gallate (2.1%), epigallocatechin gallate (6.4%), procyanidin A1 (4.2%), procyanidin A2 (5.1%), procyanidin B1 (1.4%), and procyanidin B2 (2.9%) [16]. As a natural nutrition supplement, from sport nutrition to everyday health, it exhibits superior bioavailability and numerous health benefits such as improving endurance, supporting healthy post-meal blood sugar and lipid levels, and reducing skin wrinkles and brown spots [17,18,19,20,21]. Our previous study has demonstrated that Olg mitigates age-related muscle loss by modulating molecular signaling involved in mitochondrial quality and protein turnover in senescence-accelerated mouse-prone 8 (SAMP8) mice [20]. However, the co-ingestion of Olg with BCAAs has not been determined. The present study aimed to verify the combined effect of Olg and BCAAs on sarcopenia.

## 2. Results

### 2.1. BCAAs Are Lower in Aged Sarcopenic Mice than in Young Mice

Natural aging C57BL/6 mice and SAMP8 mice have been demonstrated to present the sarcopenia phenotype at 88 weeks and 40 weeks, respectively [22]. Compared with the young mice, the concentrations of BCAAs in the gastrocnemius significantly reduced in aged sarcopenic mice (Figure 1A). It is worth mentioning that BCAAs in skeletal muscle appeared to be decreased in aged mice but significantly increased in blood serum (Figure 1B). We further hypothesized that muscle may lose the ability to transport and catabolize BCAAs in aged mice, resulting in an accumulation in circulation.

### 2.2. LAT1 Located on the Sarcolemma Is Significantly Reduced in Aged Sarcopenic Mice

The BCAA-specific transporters, such as L-Type amino acid transporter 1 (LAT1), are present on plasma membranes to import amino acids and relay nutrient signals into the cell. Elevating intracellular BCAAs, particularly leucine, activates mTOR signaling, thereby promoting muscle protein synthesis. Western blotting showed that LAT1 expressions in total lysate extraction and membrane extraction were significantly higher in the young group than in the aged sarcopenic group in C57BL6 (Figure 2A,B) and SAMP8 mice (Figure 2C,D). Via LAT1 immunofluorescent staining, more pronounced positive signals were observed at the periphery of numerous fibers in the skeletal muscle of young mice compared to those in aged sarcopenic mice (Figure 2E). The decrease in LAT1 on the sarcolemma may affect BCAA availability and nutrient signaling in aging mice.

### 2.3. LAT1 Is Essential for BCAA-Stimulated Protein Synthesis

To verify the effects of LAT1 on BCAA-stimulated protein synthesis, C2C12 myotubes were treated with scrambled or LAT1-siRNA. As shown in Figure 3A, the LAT1 mRNA level was significantly decreased by LAT1-siRNA compared with scrambled siRNA (scrambled, 1.003 ± 0.058; siRNA, 0.157 ± 0.007). In addition, the application of LAT1 siRNA did not result in a reduction in LAT2 levels (scrambled, 1.001 ± 0.037; siRNA, 1.021 ± 0.071), suggesting that the siRNA specifically targeted LAT1. LAT1-siRNA also demonstrated a significant knockdown of the endogenous LAT1 protein, as evidenced by Western blot analysis (Figure 3B and Appendix A). BCAAs (0.8 mmol/L for 16 h) significantly stimulated the phosphorylation of mTOR and p70S06K in C2C12 cells treated with scrambled siRNA. The knockdown of LAT1 did not affect the basal phosphorylation levels of mTOR; however, it did diminish the BCAA-induced phosphorylation of mTOR (Figure 3C,D). A Co-IP assay (Figure 3E,F) demonstrated that the knockdown of LAT1 in C2C12 myotubes resulted in an increased level of ubiquitination of branched-chain amino acid transaminase 2 (BCAT2). This enzyme plays a crucial role in the reversible catalysis of the initial step in the catabolism of BCAAs to branched-chain α-keto acids. In brief, BCAA deprivation reduces the activity of mTOR/p70S6K signaling, facilitates ubiquitination degradation of BCAT2, and suppresses BCAA catabolism.

### 2.4. Olg Increases LAT1 on Sarcolemma and Total BCAT2 in Skeletal Muscle of Aged Mice

We have demonstrated that Olg alleviates sarcopenia by maintaining a positive muscle protein balance and modulating mitochondrial quality in SAMP8 mice [20]. In the current study, we further investigated the role of Olg in BCAA transport and catabolism. LAT1 on the sarcolemma and the total BCAT2 in skeletal muscle were reduced in aged SAMP8 mice and reversed upon supplementation with Olg for 8 weeks (Figure 4A–D). Olg administration in aged SAMP8 mice consistently elevated BCAAs in the gastrocnemius and reduced BCAAs in the circulation (Figure 4E,F). These data suggest that Olg shunts circulating BCAAs to the skeletal muscle and improves BCAA metabolism in the skeletal muscle.

### 2.5. Olg Promotes BCAA-Stimulated Protein Synthesis by Increasing the Expression of LAT1 on the Sarcolemma

In the in vitro C2C12 myotubes, both Olg and BCAAs enhance mTOR/p70S6K signaling when used alone, and BCAAs have a stronger effect than Olg. The combination of Olg and BCAAs exhibits a slight potentiation effect (Figure 5A,B). However, blocking LAT1 through JPH203 can reverse the BCAA-stimulated enhancement of mTOR/p70S6K signaling, but does not affect Olg signaling regulation.

Hydrogen peroxide (H_2_O_2_) and palmitate (PA) can trigger various mechanisms that contribute to the pathogenesis of sarcopenia. The effect of PA on LAT1 was similar to that of H_2_O_2_ (Appendix A) but more significant. A decrease in the expression of total and membrane LAT1 in PA-treated C2C12 myotubes was restored via co-treatment with Olg (Figure 5C,D). mTOR/p70S6K signaling was significantly suppressed in C2C12 myotubes treated with PA. While BCAA supplementation alone had a minor effect on mTOR/p70S6K activity inhibited by PA, the combination of Olg and BCAAs significantly boosted this activity (Figure 5E,F). Blocked LAT1 via JPH203 can reverse the synergistic effect of the combination of Olg and BCAAs. Taken together, these data suggest that Olg increases LAT1 expression and transports BCAAs into the sarcoplasm, resulting in an increase in mTOR/p70S6K activity and protein synthesis in aged muscle.

## 3. Discussion

BCAAs are essential amino acids that can increase muscle protein synthesis through the mTOR signaling pathway, making them a favored supplement among athletes and fitness enthusiasts. BCAAs have also been studied in many muscle-wasting disorders, but the evidence regarding their effectiveness in sarcopenia is somewhat conflicting. Ma et al. reported that higher intakes of BCAAs are associated with a reduced risk of sarcopenia in a community-based observational study [23]. By contrast, Sara Ebrahimi-Mousavi et al. reported no significant association between intakes of BCAAs and odds of sarcopenia [24]. Ko et al. conducted a clinical trial on 12 pre-sarcopenic and 21 sarcopenic individuals [25]. They found that five-week BCAA supplementation may help improve muscle performance (including skeletal mass index, gait speed, and muscle strength). Still, these improvements will be obscured after 12 weeks of discontinuation. In another clinical trial reported by Mohta et al., it was concluded that supplementation of BCAAs could not ameliorate muscle mass, functional measures, or quality of life in sarcopenic patients with compensated and early decompensated cirrhosis [26]. This inconsistent evidence may result from numerous factors, including the dose of BCAAs and the overall nutritional status and health of the individual.

Aging is associated with anabolic resistance, a reduced response of muscle protein synthesis rates to typical anabolic triggers such as dietary amino acids [25,27]. This anabolic resistance may lower the availability and effectiveness of BCAAs in older adults. LAT1 is an essential component for delivering BCAAs to intracellular sensors and effector molecules associated with the mTORC1 pathway after feeding [28]. Unfortunately, the expression and function of LAT1 can be impacted during aging. Some studies showed that basal LAT1 expression was similar in young and older individuals [29,30]; however, what has been overlooked is that LAT1 must be associated with the sarcolemma to be active. Altered membrane localization of LAT1 could have a more significant impact on anabolic signals and muscle protein synthesis. In the current study, we found LAT1, especially the membrane form of LAT1, to be significantly reduced in sarcopenic conditions in both aged C57BL/6 and SAMP8 mice (Figure 2). We also showed that knockdown LAT1 via siRNA in C2C12 myotubes reduced the BCAA-induced phosphorylation of mTOR and p70S06K, indicating that LAT1 is essential in BCAA-stimulated protein synthesis (Figure 3C,D). We further speculated that the reduction in membrane LAT1 in aged skeletal muscle may lower the transmembrane flux of BCAAs (Figure 1A), accumulate BCAAs in the blood (Figure 1B), and subsequently suppress muscle protein synthesis via the down-regulation of mTOR/p70S6K signaling. BCAA supplementation may not be able to stimulate muscle protein synthesis in individuals with insufficient LAT1 on the sarcolemma. An interesting study showed that LAT1 protein increased following total-body resistance exercise training and increased more in the maltodextrin placebo group than in the leucine supplementation group [31]. However, immunohistochemistry indicated that total LAT1 in muscle fiber, but not sarcolemma LAT1, increased with training [31]. This may explain why exercise and BCAA supplementation fail to improve muscle loss in some cases.

In addition to decreased BCAA transportation resulting from the change in sarcolemma localization of LAT1, impaired BCAA catabolism in skeletal muscle promotes age-associated muscle loss. For example, the transamination of leucine to ketoisocaproic acid (KIC) is required to inhibit muscle protein degradation; lower BCAT2 would suppress muscle KIC, leading to increased muscle protein degradation [32,33]. The expression of BCAT2 was significantly decreased in 40-week-old SAMP8 mice compared with 12-week-old SAMP8 mice (Figure 4C,D). Like Lei’s study [34], we deprived BCAAs via the knockdown of LAT1 expression in C2C12 myotubes, and BCAT2 degradation via the ubiquitin proteasome pathway was observed (Figure 3E,F). Moreover, She et al. found that BCAT2 knockout mice display five-fold increased circulating BCAA levels even when they consume BCAA-free diets [33]. This is consistent with our data suggesting that circulating BCAA levels were higher in sarcopenic mice than in healthy mice.

Overcoming anabolic resistance is critical for dietary BCAA supplementation to enhance muscle mass. Olg is a low-molecular-weight polyphenol mixture of epicatechin, epigallocatechin gallate, procyanidin A2, procyanidin A1, procyanidin B2, epicatechin gallate, and procyanidin B1 [16]. It has been shown to exhibit favorable effects on anti-inflammation, suppressing intramuscular lipid accumulation and improving insulin resistance [18,19,20]. In theory, the phenolic contents of Olg will benefit muscles, particularly by contributing to positive muscle protein turnover. For example, (−)-Epicatechin presented an increase in skeletal muscle growth via inhibiting myostatin and increasing MyoD [35]. Epigallocatechin gallate stimulates muscle protein synthesis through the miR-486/AKT pathway [36]. Therefore, it is rational that Olg reduced muscle loss in diabetic db/db mice and sarcopenic SAMP8 mice in our previous studies [19,20]. Olg supplementation on healthy 24-week-old SAMP8 mice and pre-sarcopenic 32-week-old SAMP8 mice can also maintain or delay a loss of muscle mass/strength in middle and older age [37]. A study by Simona Bartova and colleagues showed that administering a grape polyphenol-rich extract decreases circulating BCAA, resulting in a protective metabolic effect against overfeeding in adult subjects [38]. Nevertheless, the role of phenolic composition in the metabolism of BCAA is largely unknown. Here, we demonstrated that Olg could increase the expression of sarcolemma LAT1 (Figure 4A,B) in aged sarcopenic SAMP8 mice. We infer accordingly that Olg may help transport circulating BCAAs into the skeletal muscle (Figure 4E,F), improve BCAA metabolism (Figure 4C,D), and activate nutrient signaling for protein synthesis. In the in vitro models, the expression of LAT1 on the sarcolemma and the signaling of mTOR/p70S6K was significantly decreased in the PA-treated C2C12 myotubes (Figure 5C–F). BCAA supplementation alone did not stimulate mTOR/p70S6K signaling significantly in PA-treated C2C12 myotubes, in contrast to the combined administration of BCAAs with Olg (Figure 5E,F). Using the LAT1 inhibitor JPH203, we found that the synergistic effect of BCAAs and Olg on mTOR/p70S6K signaling is neutralized, indicating that this synergistic effect might be attributed to the expression of LAT1 on the sarcolemma increasing due to Olg. Our results provide novel insights into the molecular mechanism through which Olg benefits BCAA transportation and catabolism. However, the combined supplementation of Olg and BCAAs in sarcopenic mice has yet to be analyzed completely since it takes a long time to develop phenotypes of sarcopenia. Animal studies are still ongoing, and more evidence will be provided to test our inference in the future.

In conclusion, the current study clarifies that Olg potentially restores the sensitivity of skeletal muscle tissue to dietary BCAAs. Combined administration of Olg and BCAAs promotes the transportation of BCAAs from the circulation to skeletal muscle via increasing LAT1 expression on the sarcolemma and further stimulates protein synthesis via mTOR/p70S6K signaling. Olg exhibits the capability to ameliorate anabolic resistance during aging and warrants further animal study and clinical research. This study provides the theoretical basis and experimental proof for the clinical application of Olg combined with BCAAs as a new therapeutic approach to sarcopenia.

## 4. Materials and Methods

### 4.1. Animals

The 88-week-old C57BL/6 mice were obtained from National Cheng Kung University (NCKU; Tainan, Taiwan). Male C57BL/6 mice were divided into two groups: young (*N* = 10, 12-week-olds) and aged (*N* = 10, 88-week-olds, with lower muscle mass and strength) [39]. The SAMP8 mice were obtained from Providence University (Taichung, Taiwan). The 12-week-old SAMP8 mice (*N* = 10, young group) and the 40-week-old SAMP8 mice (N = 20, aged group, with lower muscle mass and strength) were used in the present study. For the aged group, SAMP8 mice with pre-sarcopenic status at the age of 32 weeks were divided into two groups: SAMP8 mice (*N* = 10) fed with chow diet; SAMP8 mice administered Olg (*N* = 10) and fed with chow diet containing 200 mg Olg per kg chow diet for 8 weeks [20]. The schematic diagram for the SAMP8 animal study is shown in Appendix A. All animal experiments were approved by the Institutional Animal Care and Use Committees of NCKU and I-Shou University.

### 4.2. Cell Culture and Differentiation

C2C12 mouse myoblasts, acquired from the Bioresource Collection and Research Center (BCRC) in Hsinchu, Taiwan, were cultured in 90% DMEM medium supplemented with 10% FBS at 37 °C, 5% CO_2_. Throughout the differentiation phase, the C2C12 cells were incubated in DMEM supplemented with 2% horse serum, with the culture medium being refreshed every two days. All experimental procedures were conducted on C2C12 myotubes that had been differentiated for five days. The culture media and reagents utilized in this study were sourced from Thermo Fisher Scientific (Waltham, MA, USA).

### 4.3. Chemical

BCAAs were prepared as a mixture of leucine, isoleucine, and valine at 8 mmol/L each. Leucine, isoleucine, valine, H_2_O_2_, and palmitate (PA) were obtained from Sigma-Aldrich (St. Louis, MO, USA). Oligonol^®^ (Olg) was obtained from Amino Up Co., Ltd. (Sapporo, Japan). The selective LAT1 inhibitor, JPH203 (Nanvuranlat, KYT-0353, JPH-203SBECD), was obtained from Selleck Chemicals (Houston, TX, USA).

### 4.4. Measurement of BCAA Concentrations

BCAA concentrations in the gastrocnemius and blood serum were measured by using the BCAA Assay Kit (ab83374; Abcam, Cambridge, UK), following the manufacturer’s guidelines. Absorbance readings were taken at a wavelength of 545 nm utilizing a Microplate Reader. The BCAA concentration is determined by referencing a standard curve and normalizing it to the tissue weight or sample protein concentration.

### 4.5. Immunofluorescent Staining

Frozen soleus muscles were cryosectioned to a thickness of 6 μm onto uncoated glass microscope slides. The tissue sections were left to air dry for 10 min at room temperature, fixed with pre-cooled acetone for 10 min, and then rinsed in phosphate-buffered saline (PBS) supplemented with 0.2% Tween (PBST) to remove the fixation reagent. Nonspecific binding was blocked by incubation with 1% BSA in PBS containing 0.001% NaN_3_ at room temperature for 1 h. Primary antibodies were diluted in the blocking solution and incubated with tissue sections in a humidified environment overnight at 4 °C. Alexa Fluor goat anti-rabbit IgG 488 nm (ab150077; Abcam) served as the secondary antibody for the visualization of immunocomplexes. PBS, Tween-20, and NaN3 were obtained from Sigma-Aldrich.

### 4.6. Protein Extraction and Western Blot

Skeletal muscle tissues and C2C12 cells were washed with ice-cold PBS before being homogenized for 10 s using a hand-held homogenizer. The homogenates were then incubated on ice for 40 min in RIPA buffer supplemented with a protease inhibitor cocktail and phosphatase inhibitors (Thermo Fisher Scientific, Waltham, MA, USA). To extract plasma membrane proteins, the plasma membrane protein extraction kit (ab65400; Abcam) was utilized following the manufacturer’s guidelines. Subsequently, the protein concentrations of the tissue and cell lysates were determined using the BCA protein assay kit (5000001; Bio-Rad, Hercules, CA, USA). Equal quantities of lysate protein were subjected to SDS-PAGE, followed by electrophoretic transfer to PVDF membranes (Merck Millipore, Burlington, MA, USA). The membranes were blocked with a solution of 5% (*w*/*v*) nonfat milk powder in Tris-buffered saline containing 0.05% (*v*/*v*) Tween-20 (TBST) for 30 min at room temperature and then incubated with primary antibodies overnight at 4 °C. After being incubated with horseradish peroxidase-conjugated secondary antibody for 60 min at room temperature, the resulting bands were visualized using an enhanced chemiluminescence detection system (34577; Thermo Fisher Scientific). A list of the antibodies employed in this study can be found in Appendix A.

### 4.7. Small Interfering RNA (siRNA) Transfection

Undifferentiated C2C12 cells were seeded into 6-well plates, cultured in the growth medium until the cells reached 60% confluence, and then replaced with differentiation medium. Five-day differentiated C2C12 myotubes were transfected with an oligonucleotide for LAT siRNA (m) (cs-35796; Santa Cruz Biotechnology, Dallas, TX, USA) or scrambled control siRNA (AM4611; Thermo Fisher Scientific) using Lipofectamine 2000 (Life Technologies Japan, Tokyo, Japan) according to the manufacturer’s protocol. At 48 h after transfection, the transfected cells were harvested for further experiments.

### 4.8. Co-Immunoprecipitation (Co-IP) Assay

The Pierce™ Co-Immunoprecipitation Kit (26149; Thermo Fisher Scientific) was utilized for the Co-IP assay. Following a 48 h transfection period, cells were harvested in an IP buffer supplemented with protease inhibitors. The supernatant was then incubated with IgG, anti-BCAT2, or anti-LAT1 for one hour, after which it was combined with Protein A/G Agarose beads for an additional two hours. The agarose beads were washed with cold PBS, and the bound proteins were eluted. The ubiquitination level of BCAT2 was assessed via Western blot analysis using anti-UB, with IgG as the negative control.

### 4.9. RNA Isolation and Quantitative PCR (qPCR)

Total RNA was extracted using the TRIzol/chloroform procedure (Thermo Fisher Scientific) and reverse-transcribed into cDNAs by using the iScript™ cDNA Synthesis Kit (#1708891; Bio-Rad). Real-time qPCR was performed using TaqMan primers/probes (Slc7a5 Mm00441516_m1, Slc7a8 Mm01318971_m1, GAPDH 4352339E; Thermo Fisher Scientific) on an ABI 7900HT fast real-time PCR system. Changes in gene expression of the genes of interest were determined using the ΔΔCt method relative to the housekeeping gene GAPGH.

### 4.10. Statistical Analysis

The data obtained from the triplicate independent experiments were expressed as mean ± standard error of the mean (SEM). Statistical analysis was performed using a two-tailed unpaired Student’s *t*-test or one-way ANOVA. GraphPad Prism version 9.0 (GraphPad Software Inc., San Diego, CA, USA) was used to analyze the data, and statistical significance was established at a threshold of *p* < 0.05.

## Figures and Tables

**Figure 1 ijms-25-11549-f001:**
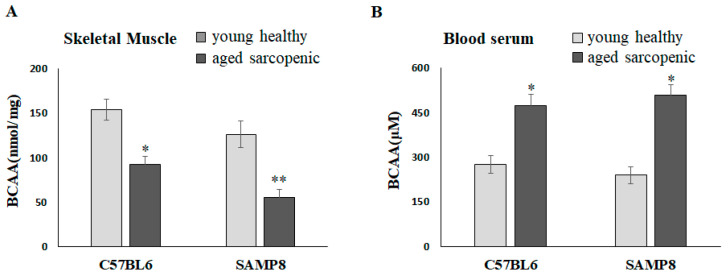
BCAAs in skeletal muscle and blood serum of mice. BCAA concentration in (**A**) skeletal muscle and (**B**) blood serum. Data represent means ± SEM (*N* = 6). * *p* < 0.05 and ** *p* < 0.01 compared to young healthy group.

**Figure 2 ijms-25-11549-f002:**
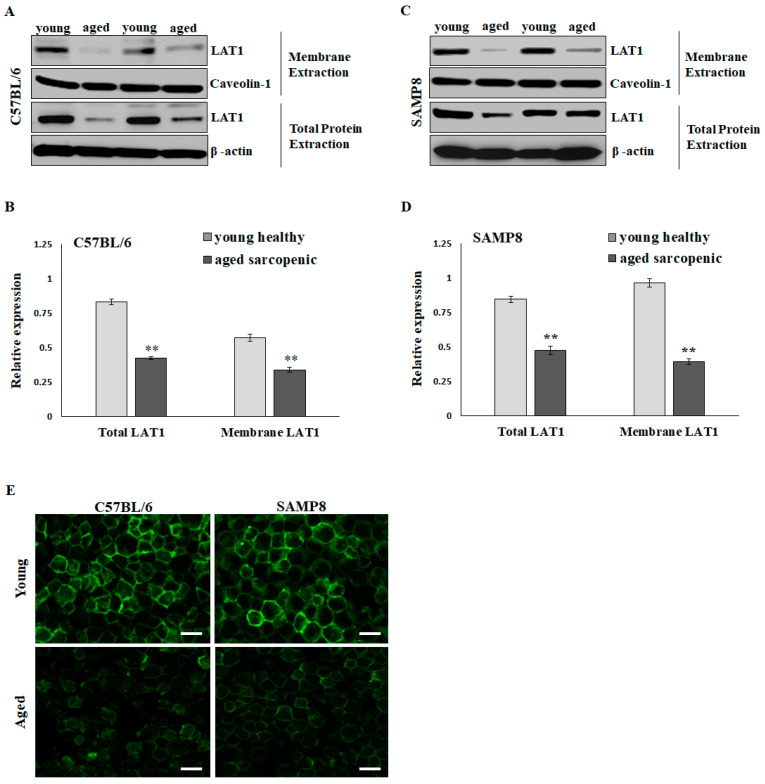
Protein expression of L-Type amino acid transporter 1 (LAT1) in skeletal muscle of mice. Representative images and quantification of total and membrane LAT1 in (**A**,**B**) C57BL/6 and (**C**,**D**) SAMP8 mice detected by immunoblots. Results are normalized to loading control and shown in histograms. Data represent means ± SEM (*N* = 6). ** *p* < 0.01 compared to young healthy group. (**E**) Immunofluorescent detection of LAT1 in young and aged mouse soleus muscle. Scale bars = 50 μm.

**Figure 3 ijms-25-11549-f003:**
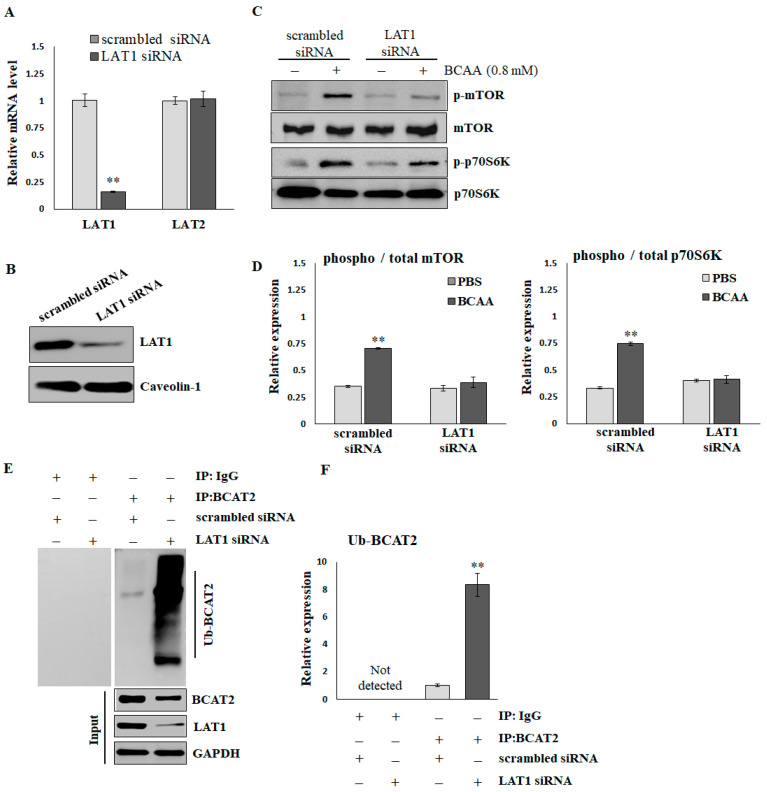
The reduction in mTOR/p70S6K signaling and ubiquitination of BCAT2 in LAT1 knockdown C2C12 myotubes. (**A**) Real-time RT-PCR and (**B**) immunoblots confirmed the knockdown efficiency. (**C**) Representative images and (**D**) quantification of p-mTOR^S2448^/mTOR and p-p70S6K^T389^/p70S6K detected by immunoblots. The results are normalized to the loading control, and then the phospho/total ratio using the normalized values is shown in the histograms. (**E**) Representative images and (**F**) quantification of ubiquitinated BCAT2 detected by Co-IP. The results are normalized to the scrambled siRNA group and shown in the histograms. The data represent the means ± SEM (*N* = 3). ** *p* < 0.01 compared to the scrambled siRNA or PBS control group.

**Figure 4 ijms-25-11549-f004:**
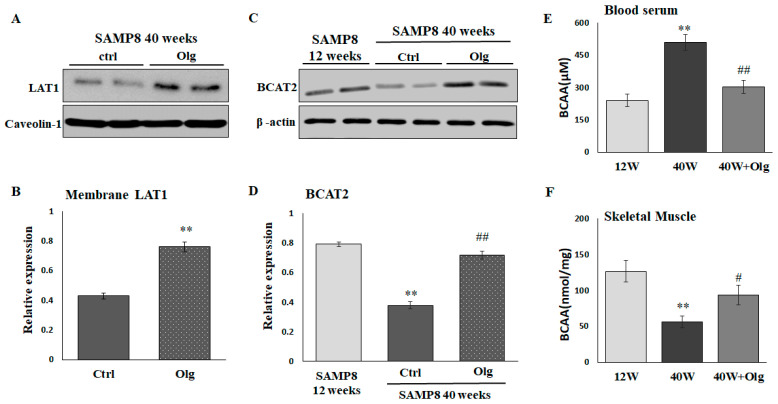
Effect of Oligonol^®^ (Olg) on protein expression involved in BCAA transportation and catabolism in SAMP8 mice. (**A**) Representative images and (**B**) quantification of LAT1 on sarcolemma detected by immunoblots. Data represent means ± SEM (*N* = 6) after normalization with loading control. ** *p* < 0.01 compared to control group. (**C**) Representative images and (**D**) quantification of BCAT2 detected by immunoblots. Data represent means ± SEM (*N* = 6) after normalization with loading control. BCAA concentration in (**E**) blood serum and (**F**) skeletal muscle. Data represent means ± SEM (*N* = 6). ** *p* < 0.01 compared to 12-week-old mice (12 w). # *p* < 0.05 and ## *p* < 0.01 compared to control diet group of 40-week-old mice (40 w).

**Figure 5 ijms-25-11549-f005:**
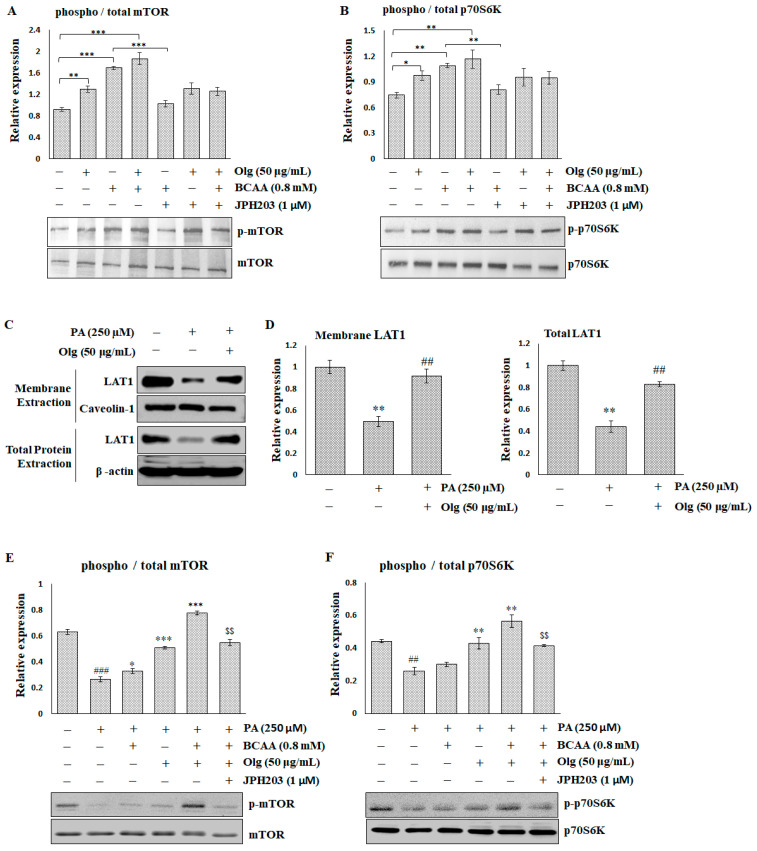
The effect of Oligonol^®^ (Olg) on sarcolemma LAT1 and mTOR/p70S6K signaling in C2C12 myotubes. Representative images and quantification of (**A**) mTOR/p-mTOR^S2448^ and (**B**) p70S6K/p-p70S6K^T389^ detected by immunoblots in the basal condition of C2C12 myotubes. The results are normalized to the loading control, and then the phospho/total ratio using the normalized values is shown in the histograms. * *p* < 0.05, ** *p* < 0.01, and *** *p* < 0.001 compared to the indicated group. (**C**) Representative images and (**D**) quantification of LAT1 on the sarcolemma and the total lysate detected by immunoblots in C2C12 myotubes treated with palmitate (PA). The results are normalized to the loading control, and then the mean ratio of the untreated group is shown in the histograms. ** *p* < 0.01 compared to the untreated control group. ## *p* < 0.01 compared to the PA-treated group. Representative images and quantification of (**E**) mTOR/p-mTOR^S2448^ and (**F**) p70S6K/p-p70S6K^T389^ detected by immunoblots in C2C12 myotubes treated with PA. The results are normalized to the loading control, and then the phospho/total ratio using the normalized values is shown in the histograms. ## *p* < 0.001 and ### *p* < 0.0001 compared to the untreated control group. * *p* < 0.05, ** *p* < 0.001, and *** *p* < 0.0001 compared to the PA-treated group. $$ *p* < 0.001 compared to PA co-treatment with BCAA and Olg. All data represent the means ± SEM (*N* = 4).

## Data Availability

The data that support the findings of this study are available from the corresponding author upon reasonable request.

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
