# Peer review of "Oligonol®, an Oligomerized Polyphenol from Litchi chinensis, Enhances Branched-Chain Amino Acid Transportation and Catabolism to Alleviate Sarcopenia"

_ijms, 2024, doi:10.3390/ijms252111549_

Round 1
Reviewer 1 Report
Comments and Suggestions for Authors
The manuscript ijms-3189387 is interesting and has the ambitious goal of demonstrating how supplementation with Oligonol® can enhances Branched-Chain Amino Acid transportation and consequently catabolism to counteract/alleviate sarcopenia, a condition unfortunately common and disabling in the elderly population.
The manuscript must be revised before it can be accepted for publication.
Major revisions
1) As requested by the editorial service, resolve the high rate of text duplicate.
2) In the introduction section, insert more specific references on the chemical composition of Oligonol®.
3) Consequently, expand the discussion on the effects of the individual components on muscle metabolism and in particular anabolism.
Minor revisions
1) Include in the text the reference to the governing structure referred to (lines 38-39).
2) In the introduction section, insert references on the molecular effects of valine and isoleucine on muscle metabolism. Don't limit yourself to the comment on leucine (inee 61-66).
3) In section 2.4 there is an error on the reference to the figure. It is probably meant figure 4 instead of 2.
4) Move figure 4 below section 2.4.
5) On line 244 remove the text highlight.
Author Response
We appreciate your precious time in reviewing our manuscript and providing valuable comments. We have carefully addressed all the comments.
Major revisions
Comment 1: As requested by the editorial service, resolve the high rate of text duplicate.
Response 1: Thank you very much for your suggestion. We have rewritten the article to reduce the duplication rate. According to the results from Turnitin, the duplication rate of our revised manuscript is 22% now. Please see the attached files.
Comment 2: In the introduction section, insert more specific references on the chemical composition of Oligonol®.
Response 2: We have added more information about the chemical composition of Oligonol® and provided corresponding references. Please see lines 80-85 of the revised manuscript.
Comment 3: Consequently, expand the discussion on the effects of the individual components on muscle metabolism and, in particular, anabolism.
Response 3: Thank you very much for your valuable advice. We also discussed the role of the phenolic contents of Olg on muscle protein turnover and BCAA metabolism. Please see lines 254-263 of the revised manuscript.
Minor revisions
Comment 1: Include in the text the reference to the governing structure referred to (lines 38-39).
Response 1: Yes, we have revised the text.
Comment 2: In the introduction section, insert references on the molecular effects of valine and isoleucine on muscle metabolism. Don't limit yourself to the comment on leucine (lines 61-66).
Response 2: Thanks for the good suggestions. We have added the molecular effects of valine and isoleucine on muscle metabolism to the introduction section. Please see lines 65-72 of the revised manuscript.
Comment 3: In section 2.4 there is an error on the reference to the figure. It is probably meant figure 4 instead of 2.
Response 3: We agree with the reviewer and have addressed this question in the revised manuscript.
Comment 4: Move Figure 4 below section 2.4.
Response 4: Yes, we have moved Figure 4 below section 2.4.
Comment 5: On line 244 remove the text highlight.
Response 5: Yes, the text highlight has been removed.
Reviewer 2 Report
Comments and Suggestions for Authors
The authors have presented through mechanistic evidence that the combination of Oligonol along with BCAA supplementation can enhance mTOR signaling and therefore protein synthesis in in vitro and in vivo models of sarcopenia.
They have also chalked out the pathway by which Oligonol enhances BCAA transport, its catabolism and downstream mTOR activation for increased protein synthesis.
Major
1. Authors have not demonstrated how co-treatment with Oligonol and BCAA translates to improvements in muscle function in sarcopenia models. It would add more impact to the current study to conduct functional studies like grip strength. Perhaps the authors have planned functional studies in a follow up manuscript.
Minor
1. In Line 240, authors discuss the effect of BCAA supplementation on anabolic resistance but do not describe in detail in the background why BCAA supplementation is detrimental. Does Oligonol co-treatment make BCAA supplementation less detrimental? Authors should clarify better.
Author Response
We would like to thank the respected reviewer 2 for his useful comments. We have tried to consider all comments and revised the manuscript based on the comments.
Major Comment
Authors have not demonstrated how co-treatment with Oligonol and BCAA translates to improvements in muscle function in sarcopenia models. It would add more impact to the current study to conduct functional studies like grip strength. Perhaps the authors have planned functional studies in a follow-up manuscript.
Response: This study provides molecular-level evidence for the benefit of Olg on BCAA transportation and catabolism; as you mentioned, the co‐ingestion of Olg with BCAAs in sarcopenic mice has not been determined. Actually, we are conducting animal experiments to provide more evidence to test our inferences. We have also revised the article so that readers can clearly understand the limitations of this study. Please see lines 274-278 of the revised manuscript.
Minor Comments
In Line 240, authors discuss the effect of BCAA supplementation on anabolic resistance but do not describe in detail in the background why BCAA supplementation is detrimental. Does Oligonol co-treatment make BCAA supplementation less detrimental? Authors should clarify better
Response: BCAAs exert muscle-building benefits, indeed, excessive BCAA supplements may harm health. We have added some introductions about the detrimental effects of BCAA and provided corresponding references. Please see lines 72-75 of the revised manuscript.
The literature review shows that the reports on the effects of BCAA supplementation in sarcopenia are controversial (the 1st paragraph of the discussion section). This inconsistent evidence may result from numerous factors, including anabolic resistance. Based on our data, we speculate that reducing membrane LAT1 in aged skeletal muscle might lower the transmembrane flux of BCAAs and subsequently suppress muscle protein synthesis via downregulation of mTOR/ p70S6K signaling. BCAA supplementation may not be able to stimulate muscle protein synthesis in individuals with insufficient LAT1 on the sarcolemma (the 2nd paragraph of the discussion section). However, Olg could increase the expression of the sarcolemma LAT1 in aged sarcopenic SAMP8 mice. We infer accordingly that Olg may help transport circulating BCAAs into the skeletal muscle, improve BCAA metabolism, and activate nutrient signaling of protein synthesis (the 3rd paragraph of the discussion section).
Combined administration of Olg and BCAAs promotes the transportation of BCAAs from circulation to skeletal muscle via increasing LAT1 expression on the sarcolemma and further stimulates protein synthesis via 264 mTOR/p70S6K signaling. To avoid misunderstanding, we have removed this sentence: "The detrimental effects of dietary BCAA supplementation in improving age-related muscle loss can be attributed to anabolic resistance in skeletal muscle" in lines 240 of the original manuscript.
Reviewer 3 Report
Comments and Suggestions for Authors
The paper of Chang et al. presents the characterization of the litchi chinensis polyphenol Oligonol© (Olg) effects on muscle BCAA metabolism during aging. The topic is of great interest and the evidence of the beneficial action of Olg against some of the defects associated with muscle aging is absolutely valuable.
Indeed, the original approach of the authors to consider the BCAA bioavailability to muscle tissue and the internalization of circulating BCAA into muscle cells as potential targets for intervention to prevent muscle waste associated with aging is very interesting.
The authors investigated the effects of Olg supplementation in combination with BCAA on the uptake and metabolism of BCAA and on the modulation of age-associated muscle phenotype; the results of the paper are promising.
However, the article is not suitable for publication in the present form. A few major issues should be addressed by the authors, with the text been revised accordingly, before considering publishable on IJMS. I also included a list of minor points, which the authors are recommended to consider; they can be found at the end of my report.
MAJOR POINTS:
1. Many of the experimental evidence come from in vitro systems. The analysis of Olg action on BCAA uptake and muscle cell signaling in vivo would add value to the study.
2. A detailed analysis of the effects of Olg treatment on young/adult muscle is missing. The authors do not reported any data on the consequence of Olg supplementation in normal non-aged muscles.
3. In general, Western blot data are presented only qualitatively, data on band quantification are not provided in the present article. A quantification of protein amount should be included.
4. Figure 2: the authors clearly showed a downregulation of LAT1 in the plasma membrane of aged muscles. Did the authors find the corresponding upregulation of LAT1 protein in the cytosolic/intracellular muscle fraction? Is there a general downregulation of LAT1 expression level? The methodology of LAT1 membrane signal quantification form the immunofluorescence images is missing. The authors should include it in the Materials and Methods section.
5. H2O2 and PA treatment are two conditions that reproduce only certain aspects of the pathophysiology of muscle aging; they do not overall imitate it. The authors’ sentence at lines 154-155 is overstated, please, rephrase it.
6. Fig. 5 As mentioned before, it would have been informative if the authors included the analysis of Olg modulation of mTOR/S6K pathway also in cell in standard condition (without treatment with PA). I suggest to add this piece of data.
Finally, I strongly encourage the authors to revise the size of text in the figures and enlarge the characters to a readable size.
MINOR POINTS:
Line 70: substitute “investigated” with the verb “understood”
Line 142: Revised the heading of paragraph 2.4. Write “…skeletal muscle of aged mice”
Line 145: use “transport” instead of “transportation”
In the Y axis of the histograms of Fig. 2-3-4-5, replace “Related” with “Relative”
Comments on the Quality of English Language
English language is overall fine, except few sentences and words that should be adjusted.
I highlighted them in my report as minor points
Author Response
We appreciate your precious time in reviewing our manuscript and providing valuable comments. We have carefully addressed all the comments.
MAJOR POINTS:
Point 1: Many of the experimental evidence come from in vitro systems. The analysis of Olg action on BCAA uptake and muscle cell signaling in vivo would add value to the study.
Response 1: Thanks for the valuable advice. This study provides molecular-level evidence for the benefit of Olg on BCAA transportation and catabolism; as you mentioned, the co‐ingestion of Olg with BCAAs in sarcopenic mice has not been determined. The animal experiments are ongoing to provide more evidence to verify our inferences. We have also revised the article so that readers can clearly understand the limitations of this study. Please see lines 274-278 of the revised manuscript.
Point 2: A detailed analysis of the effects of Olg treatment on young/adult muscle is missing. The authors do not reported any data on the consequence of Olg supplementation in normal non-aged muscles.
Response 2: Oligonol is a patented low molecular weight polyphenol extract derived from lychee fruit. Although Olg has not yet clarified its effect on young/adult muscles, it has been clinically proven to reduce fatigue, improve endurance, support healthy post-meal blood sugar and lipid levels, reduce visceral fat, and reduce skin wrinkles and brown spots. We have revised lines 85-89 in the INTRODUCTION section so that readers can know more about the health benefits of Olg, from sports nutrition to everyday health.
Our team has focused on sarcopenia research in recent years, including establishing animal models and developing new treatment strategies. We found that Olg improves protein turnover and mitochondrial mass, thereby attenuating sarcopenia in aging-accelerated mouse predisposition 8 (SAMP8) mic. Continually, we demonstrate that Olg increases sarcolemmal LAT expression in aged sarcopenic SAMP8 mice in the current study. Olg combined with BCAA may be a new therapeutic approach to sarcopenia.
Point 3: In general, Western blot data are presented only qualitatively, and data on band quantification is not provided in the present article. A quantification of protein amount should be included.
Response 3: Perhaps the layout of the figure caused misunderstanding, but in fact, we provide band quantification of protein amount in most Western Blots (except 3B and 3E). Thank you for pointing this out to me. We have improved the layout of all the figures and provided quantitative results of Figure 2B in Supplementary Figure 1. However, the ubiquitinated protein fragments are difficult to quantify, and the quantitative data has not been provided in many other published studies. If needed, we can also provide raw data from replicate experiments as supplementary information.
Point 4: Figure 2: the authors clearly showed a downregulation of LAT1 in the plasma membrane of aged muscles. Did the authors find the corresponding upregulation of LAT1 protein in the cytosolic/intracellular muscle fraction? Is there a general downregulation of LAT1 expression level? The methodology of LAT1 membrane signal quantification form the immunofluorescence images is missing. The authors should include it in the Materials and Methods section.
Response 4: We are sorry that we did not analyze the expression of LAT1 in the cytosolic fraction or whole cell. As mentioned in the DISCUSSION section (Line 216-222), some studies showed that basal LAT1 expression was similar in young and older individuals. Therefore, we focused on the membrane LAT1. We also did not quantify the results of fluorescent staining but used the Western Blot for second confirmation.
Point 5: H2O2 and PA treatment are two conditions that reproduce only certain aspects of the pathophysiology of muscle aging; they do not overall imitate it. The authors’ sentence at lines 154-155 is overstated, please, rephrase it.
Response 5: We apologize for the improper language. We have revised the sentence to “Hydrogen peroxide (H2O2) and palmitate (PA) can trigger various mechanisms that contribute to the pathogenesis of sarcopenia.” Please see lines 171-172 of the revised manuscript
Point 6: Fig. 5 As mentioned before, it would have been informative if the authors included the analysis of Olg modulation of mTOR/S6K pathway also in cell in standard condition (without treatment with PA). I suggest to add this piece of data. Finally, I strongly encourage the authors to revise the size of text in the figures and enlarge the characters to a readable size.
Response 6: Thank you very much for your valuable comments. To avoid confusion caused by too many groups, this study did not analyze the effect of Olg on the mTOR/S6K pathway in the basal condition. However, we found in previous studies that Olg increased the phosphorylation of AKT/mTOR/p70sk6, inhibited the nuclear localization of FoxO3a and NFκB, and reduced the transcription of MuRF-1 and MAFbx in vitro and in vivo. We have gone with your suggestion to modify the size of the text in the figures to a readable size.
MINOR POINTS
- Line 70: substitute “investigated” with the verb “understood”
- Line 142: Revised the heading of paragraph 2.4. Write “…skeletal muscle of aged mice”
- Line 145: use “transport” instead of “transportation”
- In the Y axis of the histograms of Fig. 2-3-4-5, replace “Related” with “Relative”
Response: Thanks for your kind reminder; we have modified all of them.
Round 2
Reviewer 1 Report
Comments and Suggestions for Authors
The authors have responded sufficiently to the concerns raised. The manuscript can be accepted for publication.
Author Response
Comment 1: The authors have responded sufficiently to the concerns raised. The manuscript can be accepted for publication.
Response 1: Thank you for your positive feedback on our manuscript. We appreciate your acknowledgment of our responses to the concerns raised.
Reviewer 3 Report
Comments and Suggestions for Authors
I carefully read the authors’ letter with their point-to-point reply. However, despite they wrote some text to discuss each of my issues, I must say that, in general, they do not satisfy my requests.
1. In vivo data are not provided.
2. The results from the analysis of healthy young animal treated with Olg are not provided.
3. Western blot bands in 3E are not quantified. Figure 3B is quantified in the supplementary material Supplementary Figure 1 despite the authors wrote in their rebuttal letter that this figure refers to main Figure 2B.
4. A quantitative evaluation of LAT1 in total cellular extract and/or cytosolic fraction is not provided. In addition, I asked for an evaluation of cytosolic LAT1 expression in the experimental conditions used by the authors in the paper, i.e. PA and H2O2 treated cells, not in young and old animals. So, they did not address my point.
5. The data on Olg effect on mTOR/S6K pathway in basal condition has not been provided. The authors mentioned other studies reporting phosphorylation of the pathway, but they did not cite these reports. In any case, I would have liked to see the effect on mTOR/Akt/S6K phosphorylation in the system used by the authors. Finally, text and axis legend of figures 3, 4, 5 and 6 are still not readable.
Figure legends of supplementary figures are missing
The topic and the results of the paper are interesting; however, few issues needed to be considered for being acceptable for publication, as I already pointed out in my first report. The authors, unfortunately, did not take into account most of my comments. With this quality of revision, I have difficulties in encouraging the publication of the manuscript even in this revised form.
Minor points:
- Line 83: correct Olga with Olg
- Line 85: correct “sport nutrition” without the s
- Line 88: the verb is missing, correct as follows: “reducing skin wrinkles and brown spots”
- Line 114: Western (blotting) should be written capitalized
- Line 156: replace skeleton with skeletal
- Line 160: replace “circulate BCAA” with “circulating BCAA”
- Line 275: replace “that benefits Olg in BCAA…” with “through which Olg benefits BCAA…”
Comments on the Quality of English LanguageMinor adjustments needed, see minor points on my report
Author Response
Response to Reviewer 3
Thank you very much for your precious suggestions and comments. After a panel discussion and study with all coauthors, we provided more precise information with reasonable explanations in the revised manuscript. The corrected parts were all in red ink color. If any questions, queries, or blurred expressions need to be further stated, don’t hesitate to ask. We will be happy to provide further information. The detail of each correction and/or explanation is listed in the following:
MAJOR POINTS:
Point 1: In vivo data are not provided.
Response 1: In the past ten years, our lab has focused on the effect of phytochemicals on age-associated diseases and has been involved in sarcopenia since 2018 [1-3]. When performing physiological analyses of sarcopenic mice (40-week-old SAMP8 and 88-week-old C57BL/6), we observed a significant decrease in the concentration of branched-chain amino acids (BCAAs) in the skeletal muscles but a substantial increase in the blood (Figure 1). Based on this finding, we hypothesized that the transportation of BCAAs in the skeletal muscles of aged mice may be lost. We further demonstrated in the current study that the L-type amino acid transporter 1 (LAT1, the BCAA-specific transporter) in aged mice is significantly reduced (Figure 2). We verified the correlation and the underlying molecular mechanism between LAT1 and BCAA-stimulated protein synthesis through the in vitro model (Figure 3). We also provided a potential role of oligomerized polyphenol (Oligonol®) in promoting BCAA transportation and catabolism to alleviate sarcopenia (Figures 4 and 5). As you mentioned, in vivo data on the combined supplement of Oligonol® and BCAAs in sarcopenic mice are not provided in the current manuscript. The in vivo animal studies are time-consuming since it takes a long time to develop phenotypes of sarcopenia. Animal experiments are still ongoing, and results are expected to be collected, analyzed, and published in the future. We also clearly explained this limitation in the DISCUSSION, please see lines 287-290 of the revised manuscript.
Point 2: The results from the analysis of healthy young animals treated with Olg are not provided.
Response 2: Olg is safe as a food or dietary supplement and has been clinically proven to improve health and physical performance (please see lines 85-88 of the revised manuscript). Based on our unpublished data obtained from student thesis in our lab [4], Olg supplementation on healthy 24-week-old SAMP8 and pre-sarcopenic 32-week-old SAMP8 can maintain or delay loss of muscle mass/strength in the middle and older age (please see attached file). We have added the information in the revised manuscript in lines 269-271.
Point 3: Western blot bands in E are not quantified. Figure 3B is quantified in the supplementary material Supplementary Figure 1 despite the authors wrote in their rebuttal letter that this figure refers to main Figure 2B.
Response 3: We are sorry for our inadvertent. We provide quantitative results for Figure 3B in Supplementary Figure 1 and quantitative results for Figure 3E in revised Figure 3F.
Point 4: A quantitative evaluation of LAT1 in total cellular extract and/or cytosolic fraction is not provided. In addition, I asked for an evaluation of cytosolic LAT1 expression in the experimental conditions used by the authors in the paper, i.e. PA and H2O2 treated cells, not in young and old animals. So, they did not address my point.
Response 4: The expression of total LAT1 in both animal and C2C12 cells has been provided in the revised manuscript. Please see the revised Figure 2A-D, Supplementary Figure 2, and Figure 5C-D. We hope these adjusted data will address your point. Thank you for the suggestion again.
Point 5-1: The data on Olg effect on mTOR/S6K pathway in basal condition has not been provided. The authors mentioned other studies reporting phosphorylation of the pathway, but they did not cite these reports. In any case, I would have liked to see the effect on mTOR/Akt/S6K phosphorylation in the system used by the authors.
Response 5-1: The data is now in revised Figure 5. In basal conditions, both Olg and BCAAs enhance mTOR/p70S6K signaling when used alone, and BCAAs have a stronger effect than Olg. The combination of Olg and BCAAs exhibits a slight potentiation effect. However, blocking LAT1 through JPH203 can reverse the BCAA-stimulated enhancement of mTOR/p70S6K signaling but does not affect Olg signaling regulation (Figure 5A-B with description in lines 173-177).
Point 5-2: Finally, the text and axis legend of figures 3, 4, 5, and 6 are still not readable.
Response 5-2: Thank you for pointing this out to us. We have tried our best to resolve this issue, and we will confirm with the editorial office that the figures meet their requirements.
Point 6: Figure legends of supplementary figures are missing.
Response 6: We have added figure legends of supplementary figures in the revised version. Please see the field titled “Supplementary Figure_R2”.
The topic and the results of the paper are interesting; however, few issues needed to be considered for being acceptable for publication, as I already pointed out in my first report. The authors, unfortunately, did not take into account most of my comments. With this quality of revision, I have difficulties in encouraging the publication of the manuscript even in this revised form.
Response: Again, thank you for your valuable feedback on our manuscript. We appreciate your acknowledgment of the topic and the exciting results. We have tried addressing your concerns and incorporating relevant materials in our revisions. We would appreciate it if you could take another look at the updated manuscript. Your insights are invaluable, and we hope to meet the publication standards.
MINOR POINTS
- Line 83: correct Olga with Olg
- Line 85: correct “sport nutrition” without the s
- Line 88: the verb is missing, correct as follows: “reducing skin wrinkles and brown spots”
- Line 114: Western (blotting) should be written capitalized
- Line 156: replace skeleton with skeletal
- Line 160: replace “circulate BCAA” with “circulating BCAA”
- Line 275: replace “that benefits Olg in BCAA…” with “through which Olg benefits BCAA…”
Response: We apologize for our carelessness. We have corrected these typos and grammar, as you suggested. Thank you for your help with our manuscript.
REFERENCES
- Chang, Y.C., et al., Oligonol Alleviates Sarcopenia by Regulation of Signaling Pathways Involved in Protein Turnover and Mitochondrial Quality. Mol Nutr Food Res, 2019. 63(10): p. e1801102.
- Chang, Y.C., et al., Resveratrol protects muscle cells against palmitate-induced cellular senescence and insulin resistance through ameliorating autophagic flux. J Food Drug Anal, 2018. 26(3): p. 1066-1074.
- Liu, H.W., et al., Dysregulations of mitochondrial quality control and autophagic flux at an early age lead to progression of sarcopenia in SAMP8 mice. Biogerontology, 2020. 21(3): p. 367-380.
- Tseng, W.-T., Oligonol, a Low-Molecular Weight Polyphenol Derived from Lychee, attenuates muscle atrophy in senescence-accelerated mouse prone 8 (SAMP8) mice. 2018, National Cheng Kung University.

Round 3
Reviewer 3 Report
Comments and Suggestions for Authors
The authors revised their manuscript adding a few experimental data that I asked for and reply to my second-round comments. I appreciated the work they have done throughout the revision process and their effort in satisfying my requests.
The manuscript has been improved since the first submission, and it can be considered satisfactory.
I approve its publication in the IJMS journal.